# Interaction of Thymine DNA Glycosylase with Oxidised 5-Methyl-cytosines in Their Amino- and Imino-Forms

**DOI:** 10.3390/molecules26195728

**Published:** 2021-09-22

**Authors:** Senta Volkenandt, Frank Beierlein, Petra Imhof

**Affiliations:** 1Department of Physics, Freie Universität Berlin, Arnimallee 14, 14195 Berlin, Germany; senta.volkenandt@fau.de; 2Computer Chemistry Centre, Department for Chemistry and Pharmacy, Friedrich-Alexander University (FAU) Erlangen Nürnberg, Nägelsbachstrasse 25, 91052 Erlangen, Germany; frank.beierlein@fau.de; 3Erlangen National High Performance Computing Center (NHR@FAU), Martensstraße 1, 91058 Erlangen, Germany

**Keywords:** thymine DNA glycosylase, amino imino tautomers, oxidised methyl-cytosine, molecular dynamics simulations, alchemical free energy calculations

## Abstract

Thymine DNA Glycosylase (TDG) is an enzyme of the base excision repair mechanism and removes damaged or mispaired bases from DNA via hydrolysis of the glycosidic bond. Specificity is of high importance for such a glycosylase, so as to avoid the damage of intact DNA. Among the substrates reported for TDG are mispaired uracil and thymine but also formyl-cytosine and carboxyl-cytosine. Methyl-cytosine and hydroxylmethyl-cytosine are, in contrast, not processed by the TDG enzyme. We have in this work employed molecular dynamics simulations to explore the conformational dynamics of DNA carrying a formyl-cytosine or carboxyl-cytosine and compared those to DNA with the non-cognate bases methyl-cytosine and hydroxylmethyl-cytosine, as amino and imino tautomers. Whereas for the mispairs a wobble conformation is likely decisive for recognition, all amino tautomers of formyl-cytosine and carboxyl-cytosine exhibit the same Watson–Crick conformation, but all imino tautomers indeed form wobble pairs. The conformational dynamics of the amino tautomers in free DNA do not exhibit differences that could be exploited for recognition, and also complexation to the TDG enzyme does not induce any alteration that would indicate preferable binding to one or the other oxidised methyl-cytosine. The imino tautomers, in contrast, undergo a shift in the equilibrium between a closed and a more open, partially flipped state, towards the more open form upon complexation to the TDG enzyme. This stabilisation of the more open conformation is most pronounced for the non-cognate bases methyl-cytosine and hydroxyl-cytosine and is thus not a likely mode for recognition. Moreover, calculated binding affinities for the different forms indicate the imino forms to be less likely in the complexed DNA. These findings, together with the low probability of imino tautomers in free DNA and the indifference of the complexed amino tautomers, suggest that discrimination of the oxidised methyl-cytosines does not take place in the initial complex formation.

## 1. Introduction

Thymine DNA glycosylase (TDG) is an enzyme of the base excision repair (BER) system that recognises and excises the nucleobase of a number of damaged or mispaired nucleotides. In addition to the removal of the name-giving thymine in G:T mispairs, TDG has been reported to operate on forms of oxidised methyl-cytosine, products of the ten-eleven translocation (TET) methyl-cytosine dioxygenase that transforms 5-methyl-cytosine (MC) through step-wise oxidation into 5-hydroxymethyl-cytosine (HMC), 5-formyl-cytosine (FC), and 5-carboxylcytosine (CAC) [1,2,3]. Whereas HMC and MC are not processed by the glycosylase enzyme, the higher oxidised forms—FC and CAC—are recognised and expelled by TDG, and ultimately, by other enzymes in the base excision repair pathway, replaced by unmethylated cytosine [1,4,5].

Crystal structures of TDG glycosylases complexed to lesioned DNA [6,7,8] show the mispaired or damaged bases flipped out of the helical DNA duplex into the enzyme’s active site. Base extrusion is thus an important step in the base excision process and one possibility along a multi-step interrogation pathway to discriminate target bases from non-cognate ones [9,10]. The base flip has been suggested by simulations to follow different dynamics for FC and CAC than for thymine, and an active role of the TDG enzyme in promoting base extrusion has been shown [11].

Besides discrimination of the cognate and non-cognate methyl-cytosine forms at the stage of base flipping, substrate specificity can at last be achieved at the chemical step of glycosidic bond cleavage between the C1’-atom of the sugar and the N1-atom of the methyl-cytosine base. The removal of CAC has been shown to be acid catalysed, ruling out HMC and MC as substrates since these bases have no proton acceptor groups. Excision of FC, on the other hand, does not require acid catalysis [4] but appears to rely on FC to be a better leaving group than HMC or MC [12,13,14]. Quantum chemical calculations on nucleotide models suggest that differences in the ‘inherent chemistry of the modifications’ compared to MC lead to lower barriers of the glycosidic hydrolysis and thus higher activity of TDG [15,16,17]. One such chemical difference has been described by shorter N-glycosidic bond lengths in the reactant transition state and another by the leaving group ability of the base [18]. Calculations of thymine excision by the TDG enzyme find an active role for a histidine residue, His151, in proton shuttling to and from the leaving base, a mechanism that is not conceivable with cytosine and methyl-cytosine bases [19]. Calculations of the excision mechanism of FC in TDG, in contrast, do not support the need for proton shuttling by His151 and do not provide further suggestions of substrate discrimination at the chemical step [20].

Biochemical DNA binding data show binding to C, MC and HMC to be significantly weaker than binding to DNA with substrate bases [4,14]. It is therefore conceivable that the recognition by the glycosylase has taken place already, upon binding to the damaged DNA, forming stronger or at least different interactions with the two cognate forms of oxidised methyl-cytosine, FC and CAC, than with the non-cognate forms, HMC and MC, respectively. Figure 1 summarises possible steps in the protein binding, base recognition, and base excision mechanisms by which TDG could discriminate target and non-target bases.

In contrast to the recognition of mispaired thymine, the deamination product of methyl-cytosine (or uracil, the deamination product of cytosine), which is likely detected due to the local deformation of the DNA at the lesion site [21,22], none of the aforementioned oxidised methyl-cytosine forms appears to exhibit an altered conformation of the DNA in solution [23,24] that can easily be recognised by the repair enzyme TDG. Experimental and simulation studies, however, report distinct structural alteration of 10 bp long DNA with two XC bases, one FC or CAC on either strand, compared to B-DNA [25]. Moreover, larger fluctuations of FC and HMC have been observed in molecular dynamics (MD) simulations and larger fluctuations of FC:G pairs have also been confirmed experimentally [26]. Base pair opening, which can be understood as the first step of base extrusion, has been probed by NMR, using imino proton exchange rates as a marker. Imino protons are more accessible and thus the exchange rates are faster, if the G:XC pair is in a (partially) open conformation and/or the base is (partially) flipped out of the DNA helix. The measured rates, though different for the different oxidised forms, do not show a trend that correlates with TDG activity [27]. In particular, CAC does not alter DNA flexibility and does not exhibit greater base pair motion (opening) or faster imino proton exchange [24,27]. ${}^{13}$C-NMR vs. pH-titration experiments show a much lower N3-pK_a_ for FC and CAC than for HMC, which has been explained by the electron-withdrawing properties of the formyl and carboxyl group. An increased N3 acidity would then correlate with weakened hydrogen bonding and reduced base pair stability, explaining the observed lower melting temperatures for FC and CAC compared to MC [28].

It has also been suggested that the higher oxidised forms of methyl-cytosine—FC and CAC—could be recognised because of their higher propensity, compared to HMC and MC, to form imino tautomers [4,7]. Such imino tautomers would predominantly form so-called wobble pairs (see Figure 2), which resemble the mispairs formed by uracil, G:U, and thymine, G:T. Calculations of amino and imino tautomers of the isolated nucleobases FC and CAC in the gases phase and in an implicit water model show a clear preference for the amino forms [6]. Moreover, NMR experiments [27] and 2D-IR spectra accompanied by density functional theory calculations [28] find the amino forms of free DNA in water to be predominant.

Yet, the situation might be different in the protein environment, that is, with the TDG enzyme complexed to the DNA, an imino form might be stabilised by the enzyme and/or DNA with one of the oxidised forms, amino or imino, interacting more or less favourably than the others. In this paper, we therefore investigate DNA carrying one of the different possibilities of G:XC pairs at a time, namely guanine paired to one of the differently oxidised forms of methyl-cytosine in their amino (XC = MC, HMC, FC, or CAC) or imino (XC = IMC, IHC, IFC, or ICC) tautomeric forms, respectively. By means of molecular simulations, we compare these DNA systems, in free form and complexed to TDG, so as to explore the differences in conformational dynamics and protein-DNA interactions between the different oxidised forms and between amino and imino tautomers.

## 2. Methods

We modelled DNA with different modifications of a G:XC pair as amino tautomers, XC = CAC, FC, HMC, MC and imino tautomers XC = ICC, IFC, IHC, IMC. The starting coordinates of the uncomplexed DNA in sequence CATCGCTCA**XC**GTACAGAGC have been taken from the PDB [29,30] structure 6U17 [31]. For the complex of DNA (GCTCA**XC**GTACA) with thymine DNA glycosylase we used the crystal structure with PDB code 2RBA [6]. This structure contains a 2:1 complex of the catalytic domain of human TDG (residues 111–308) with one protein bound to an abasic site analog and the other one bound to a non-cognate site with a central G:C pair. We removed the protein bound to the abasic site and from the DNA we kept only the part that is complexed to the second other protein. The different modifications, XC = CAC, FC, HMC, MC and imino tautomers XC = ICC, IFC, IHC, IMC, have been build by adding a (oxidised) methyl group to the 5C of the cytosine base of the central G:C pair. Molecular dynamics simulations of the models were performed with Amber 18 [32] and Amber 20 [33] using pmemd.cuda, following a protocol established previously [34,35,36,37]. The DNA part of the system was described by the parmbsc1 [38] force field and the protein by ff14SB [39]. Both free and complexed DNA were solvated with TIP3P [40] water and sodium counter ions were added as well as NaCl at a concentration of 150 mM [41].

5-Methylcytosine (MC) and its oxidised derivatives 5-hydroxymethylcytosine (HMC), 5-formylcytosine (FC) and 5-carboxylcytosine (CAC) and their imino tautomers ICC, IFC, IHC and IMC (one hydrogen moved to N3 from N4, see Figure 2) were parameterised following a protocol established previously [35,36,37,42,43]. Therefore, only a short summary is given here. We used RESP [44,45] charges for the modified bases; missing parameters of the bases were amended using values from GAFF [46,47] or parmbsc1/ff14SB [38,39].

After initial geometry optimisation and 500 ps heating to 298 K in an NVT ensemble, for each of the systems, three independent runs were performed for 600 ns. These production runs were performed with Langevin dynamics in an NPT ensemble at 1 bar and 298 K, using a time step of 2 fs, SHAKE [48,49] on all bonds involving hydrogen and periodic truncated octahedral boxes (box dimensions approx. 92 Å), a non-bonding cutoff of 10 Å, and particle mesh Ewald for the treatment of electrostatic interactions. Watson–Crick distance restraints were imposed on the DNA termini of the free DNA (20 kcal·mol ${}^{-1}\cdot$Å${}^{-2}$, allowing ±0.1 Å movement from the equilibrium bond distance) to prevent fraying of the DNA termini [50]. For the protein-DNA complexes we used slightly different distance restraints with a force constant of 2 kcal·mol ${}^{-1}\cdot$Å${}^{-2}$ on the DNA termini, but the same as in [51], which are in accordance with B-DNA geometry. Not only were the distances between the heavy atoms of the Watson–Crick H-bonds of the DNA termini restrained but also the C1’-C1’ distance of them and the ones shifted by one base pair in the 3’ or 5’ direction (for upper and lower bounds see Supporting_Table S1 of [51]).

Only the last 500 ns were used for analysis. Cpptraj [52] from the AmberTools suite, vmd 1.9.3 [53] and Curves+/Canal [54,55] were used for further analyses of the systems’ fluctuations, hydrogen-bond interactions and the DNA conformation. A hydrogen bond was defined based on geometric criteria, that is, a donor–acceptor distance not larger than 3.2Å and a donor-hydrogen-acceptor angle deviating from linearity by not more than 42${}^{\circ }$. A flip angle, describing how much the XC base is in an intrahelical or extrahelical state, is defined as the pseudo dihedral formed by the XC base, the sugar of the XC nucleotide, the sugar of the next nucleotide downstream and the next base and its complementary base, a definition we have used previously [21,22].

Relative binding free energies of the complexes of the DNA oligonucleotides containing the amino- and imino-tautomers of the four (oxidised) variants of 5-methylcytosine with TDG were obtained using the thermodynamic cycle shown in Figure 3. The perturbations were performed with Amber 20 [33] pmemd.cuda following a dual-topology thermodynamic integration (TI) approach [56,57,58,59]. The amino tautomers of CAC, FC, HMC and MC were perturbed into their imino tautomers ICC, IFC, IHC and IMC using a lambda coordinate of 21 windows (0.00, 0.05, …, 0.95, 1.00), both in the bound state (complex with TDG) and the free state (solvated in water) [56]. A van-der-Waals and electrostatic soft core potential with Amber 20 default soft core parameters was used, the soft core regions are indicated in Figure 4.

The perturbation free energies $\Delta G_{pert}^{bound}$ and $\Delta G_{pert}^{free}$ (Scheme in Figure 3) were obtained from the free energy gradients by trapezoidal numerical integration. Starting structures for the TI simulations were taken from the MD simulations of the unperturbed amino forms of CAC, FC, HMC and MC after 10.5 ns equilibration. Again, Watson–Crick distance restraints on DNA termini (see above) were employed to prevent fraying of the DNA termini.

After initial geometry optimisation, each lambda window was heated to 298 K during 200 ps NVT with weak Cartesian restraints (5 kcal·mol ${}^{-1}\cdot$Å${}^{-2}$) on non-hydrogen DNA/protein atoms, followed by 200 ps NPT equilibration without restraints. An integration time step of 1 fs was used, with SHAKE [60] constraints on all bonds involving hydrogen except the perturbed residues (in addition to SHAKE being removed between bonds containing one common and one unique atom). A Monte Carlo barostat was used for pressure (1 bar) control. All other simulation parameters were chosen as suggested in the Amber TI tutorial [56]. Each lambda window was simulated for 30 ns, of which the last 20 ns were used for integration and each perturbation was repeated 2 times so as to generate 3 runs of each perturbation simulation.

Values reported in this work are the mean calculated from averaging over the three independent runs of the respective simulations and errors are estimated as the standard deviation from the mean.

## 3. Results

### 3.1. Comparison of Free and Complexed DNA

#### 3.1.1. DNA Conformation

In the free DNA, the axis bend is generally slightly larger for the imino forms of XC than for the amino forms. In contrast, in the complexed DNA, there is an increased axis bend at the location of the G:XC pair and its neighbouring G:C base pair, independent of the oxidation state or the tautomeric form of the XC. Such a localised bend is not observed in the free DNA (see Figure 5).

Differences in the other DNA parameters between the different oxidised forms and/or between amino and imino tautomers are also rather localised, affecting only the G:XC pair itself or its direct neighbours (see Appendix A). Base step parameters shift, rise, roll and twist show differences between the amino and imino tautomeric forms of the XC, located at the G:XC step. In the complexes, these differences are, however, less pronounced (see Appendix A).

Shear, stretch and opening angle show clear differences between the amino and imino tautomers of the XC, with larger displacements in the imino forms, reflecting their wobble conformation (see Appendix A). Closer inspection of the probability distribution and free energy profile of the stretch displacement (Figure 6) and the opening angle (Figure 7) at the G:XC lesion reveal a two-state scenario for the imino tautomers. The more populated state with lower free energy has stretch and opening values comparable to those observed for the only state in the amino tautomers. The second, less populated state with consequently higher free energy, is shifted towards higher stretch displacements and higher opening angles, indicating an even more deformed, partially open base pair. Whereas in the free DNA, the relative free energy of the second state is comparable for all forms of oxidised methyl cytosine, the complexed DNA exhibits a trend for both stretch and opening angle with relative free energies decreasing in the order ICC > IFC > IHC > IMC. This suggests a more open conformation to be more favourable at the lower oxidation level. The comparably large errors in the free energy profiles of the complexed imino tautomers, as opposed to very small errors in the amino tautomers, further suggest higher conformational freedom for the imino systems. Note, however, that the errors reflect differences between the individual simulation runs and not the fluctuations within one single simulation. As the time series of the opening angle and flip angle show (see Appendix A), there are fluctuations between the two conformational states as well as individual simulations that are predominantly in one or the other state.

Similar to the opening angle, the flip angle exhibits two states in the imino tautomers but only one state in the amino tautomers of the G:XC pair (see Figure 8). In line with the observations made for the opening angle, the second, more flipped state has a higher free energy than the closed/unflipped state. Yet again, the relative free energy values are comparable for the four forms of oxidised methyl-cytosine in free DNA, whereas in the complex, the second state is the more probable the lower the oxidation level is. The complexed imino forms IMC and IHC show a higher probability of a larger flip angle than the imino forms IFC and ICC.

#### 3.1.2. Hydrogen-Bond Interactions

The Watson–Crick pairing of the amino forms and the wobble conformation of the imino forms (see Figure 9) are clearly mirrored in the hydrogen bonds between the XC and the complementary guanine base. In the amino forms, three hydrogen bonds are formed almost throughout the entire simulation time (probabilities of 0.90–0.98, see Table 1 and Figure 10). In the imino forms, the hydrogen bond donated by the N1 atom of the guanine base is accepted by the O6 atom of the XC, and a second hydrogen bond is formed between the now donating N3 atom of XC and the O6 atom of the guanine. This second hydrogen bond has, however, a probability reduced to only 0.73–0.83 as opposed to hydrogen bonds with more than 0.9 probability in the Watson–Crick conformations. The remaining probability (∼0.2) is spent for hydrogen bonds between the N2 atom of the guanine and the O2 atom of the XC, that can be regarded as a remnant of the Watson–Crick conformation and incomplete wobble.

In the complexed DNA, the situation is unaltered for the amino forms, that is, three hydrogen bonds with a probability larger than 0.9 are observed. Of the two hydrogen bonds in the wobble pairs of the imino forms, there is even higher probability (∼0.4 for FC and CAC and 0.6–0.7 for HMC and MC, respectively) for the residual hydrogen bond between the N2 atom of the guanine and the O2 atom of the XC. In addition, the probabilities for the hydrogen bond between the N3 atom of the XC and the O6 atom of the guanine in the wobble pairs differ between the different oxidised forms with increasing probability from MC over HMC and FC to the highest one in CAC. The second hydrogen bond, between the N1 atom of the guanine and the O2 atom of the XC, has, within error, comparable probabilities for the four differently oxidised G:XC pairs, but even here the higher oxidised forms, FC and CAC, exhibit the highest probabilities.

Another indicator and probably also stabilising element of the wobble conformation and the partially open conformation of the imino G:XC pairs is the high probability (∼0.6, see Table 2) for observing a water molecule bridging the guanine and the oxidised methyl cytosine in the free DNA. No such water-bridged hydrogen bond is observed in the amino forms. Upon complexion to the TDG protein, the probability of water-mediated hydrogen bonds is still zero for the amino forms, and for the imino forms this probability drops to ∼0.5 and for ICC even to ∼0.35 (see Table 2).

Analysis of interactions between water and the bases of the G:XC lesion, in terms of hydrogen bonds, shows a few obvious differences such that HMC and IHC are the only systems in which the XC base is a hydrogen bond donor to a water molecule and hydrogen bonds with oxygen atom O16 can only occur in the CAC and ICC systems since only the carboxyl group has this second oxygen atom. Likewise, MC and IMC lack oxygen atom O15 and therefore cannot accept a hydrogen bond from a water molecule with the methyl group (see Figure 2). It is interesting to note, however, that O15 acts also as hydrogen bond acceptor in the amino tautomer HMC, in free and complexed form, but only in the free DNA also in the imino tautomer IHC (see Table 3).

The N4 atom can accept hydrogen bonds with water only in the imino tautomers and does so with high probability in free and complexed DNA for all systems but ICC. In this system it is likely due to the (lost) competition with two carboxyl oxygen atoms. None of the amino tautomers forms a hydrogen bond with water in which the amino group is the donor, despite only one of its hydrogen atoms being involved in base pairing with the guanine. In contrast, the O6 atom of the guanine accepts a hydrogen bond from a water molecule (in addition to the one from base pairing with XC) with a very high probability in almost all systems, except for ICC, where this hydrogen bond is not observed, neither for the free nor for the complexed DNA (see Table 3). The probabilities for the other polar atoms of the guanine base are almost unaltered between tautomers and different oxidation levels. Only the hydrogen bond between the N3 atom and water molecules, which has a probability of ∼0.6 in all free DNA systems and all complexed amino tautomers, is lacking in the complexes of the imino tautomers ICC and IHC (see Table 3).

Regarding the XC base, the most striking difference between the systems investigated in this work is the hydrogen bond probability of the O2 atom with water. This atom is hydrogen bond acceptor to the guanine base in the Watson–Crick conformation of the amino tautomers as well as in the wobble pair conformation of the imino tautomers (see Figure 2 and Table 1). Yet, it accepts a second hydrogen bond from a water molecule with high probability only in the amino tautomers and in the ICC imino tautomer of free DNA. In the complexed DNA this hydrogen bond is mostly absent, except for the CAC system that exhibits a moderate probability for such a hydrogen bond, albeit with large error (see Table 3).

#### 3.1.3. Fluctuations

Fluctuations of the DNA carrying different forms of XC are comparable between amino and imino forms in the free DNA and in the complex, respectively. Ignoring the terminal base pairs, however, the complexed DNA exhibits significantly less fluctuations than the free DNA which, in particular in the region of the lesion, fluctuates up to 2 Å compared to fluctuations of only ∼1 Å otherwise (see Figure 11). The complexed imino forms of the oxidised methyl-cytosines exhibit a larger error in their fluctuations than the corresponding amino forms, which is most pronounced for the lesion XC and its complementary guanine residue.

### 3.2. TDG-DNA Interactions

#### 3.2.1. Hydrogen-Bond Interactions

There are no hydrogen bonds between residues of the TDG protein and the XC base that show a probability of at least 0.5. The only direct contacts between the XC base and the protein are by Lys201 and the oxygen atom(s) of the (oxidised) methyl group of XC, but with rather low probability: The amino and imino tautomers of carboxyl-cytosine, CAC and ICC, have a probability of ∼0.2 to form such a hydrogen bond, the same holds for the imino tautomer IHC (see Table 4). In all cases, however, the error is about as large as this mean value.

All other hydrogen bonds formed between the protein and the DNA are either with the backbone of the XC nucleotide, but with rather low probability (see Table 4) or with residues of the DNA other than the G:XC pair (see Table 5). Hydrogen bonds of Arg275 are observed with the backbone of the next neighbours of the G:XC pair, that is with the O2 atom of the T18. This hydrogen bond is observed in all systems, albeit with large fluctuations, except for the tautomer ICC for which G19 hydrogen-bonds to Arg275 (see Table 5). With the other DNA strand, there are highly probable hydrogen bonds with Lys232 and the imino tautomers, one between T8 and and the backbone of Lys232 (except for IFC) and another one with lower probability between A9 and the NZ atom of Lys232’s side chain. This latter hydrogen bond has higher probability in the lower oxidised forms, IHC and IMC (see Table 5) and is present only with the amino tautomer of HMC.

#### 3.2.2. Protein-XC Distances

Differences in the interactions with the TDG protein between the different XC bases, amino and imino tautomers and different oxidation levels, are also manifested by the accommodation of the XC base in the complex.

Figure 12, Appendix A, depict the probability distributions of the distances of the XC base to the protein residues Lys201 and Pro202, respectively. Lys201 has a low probability to form hydrogen bonds to the oxygen atom of the oxidised methyl-cytosines (see Table 4) and the distance distributions (Appendix A) show that Lys can come close enough, in particular in the CAC system, but is mainly at a distance of ∼5–7 Å in the amino tautomers. For the imino tautomers, the probability of shorter Lys201-XC distances increases slightly, due to the more open or wobble conformation. The same holds for the distance to the N4 atom of the XC base (see Figure 12). This latter distance also allows comparison with the MC and IMC systems which show a similar trend of shorter distances in the imino tautomer. The amino tautomer, MC, has the largest distance between its N4 atom and Lys201 (∼9 Å, see Appendix A), indicating that with the lacking oxygen atom there is also a favourable interaction missing. In the imino tautomers, in contrast, there is a possible, though not fully realised, interaction with the N4 atom as a putative hydrogen bond acceptor if the imino proton, NH41 atom, is oriented towards the N3 atom of the base, and away from Lys201 (see Figure 13). As the probability distribution of that dihedral angle shows (see Appendix A), such conformations, corresponding to a HN41-N4-C4-N3 dihedral angle of ∼0${}^{\circ }$ (as opposed to ∼$180^{\circ }$ when pointing away from the N3 atom), are observed for free and complexed DNA of IHC and IMC, but not for the higher oxidised forms, IFC and ICC, neither in free nor in complexed DNA. Short Lys203-XC distances in the IFC and ICC systems are thus favoured by interactions with the oxygen atom(s) of the formyl and carboxyl group, respectively.

For the imino tautomers favourable interactions with Lys201 are facilitated by a more open conformation of the G:XC pair. The two-state nature of the imino conformations is not reflected in the distance distributions to Lys203, likely due to the flexibility of this residue. Analysing the distances to the rather rigid neighbouring Pro202, though, shows a bimodal distribution for the imino tautomers, and even for the HMC amino tautomer. The amino tautomer MC, in contrast to the other amino tautomers with broader Pro202-XC distance distributions, is dominantly at distances of ∼12–14 Å from Pro202, corresponding to the large distances to Lys201 observed for this system.

#### 3.2.3. Relative Binding Affinities

Alchemical perturbation and thermodynamic integration reveal that the amino and imino tautomer of the highest oxidised form, carboxyl-cytosine, show similar binding affinities (see Table 6). For hydroxymethyl-cytosine, complexation of the imino tautomer is slightly favoured over complexation of the amino form. The other two oxidation forms, formyl- and methyl-cytosine exhibit, within errors, the same difference in binding affinities. In these systems complexation of the amino tautomer is by ∼2 kcal/mol more favoured than binding of the imino tautomers to TDG.

## 4. Discussion

The conformational dynamics and the interactions with the protein do not exhibit significant differences between the different oxidised forms of 5-methyl-cytosine in their amino tautomers. However, the imino tautomers of all G:XC pairs exhibit significant conformational differences compared to their amino counterparts. As anticipated, the amino tautomers are in a Watson–Crick conformation whereas the imino tautomers form wobble pairs with fewer and shifted hydrogen bonds between the XC and the guanine base. Such differences can in principle be exploited for recognition by the TDG protein and it has also been suggested that extrusion (flipping) of an XC base in a wobble pair requires less energy than from a Watson–Crick pair [7]. The wobble conformations, moreover, exhibit a second, less favourable conformational state, that corresponds to a partially open and partially flipped state, as has been observed earlier for mispaired thymine [21,22]. For the mispair, this second state is stabilised upon complexation to the TDG protein, in contrast to G:C and G:MC (in amino form) which remain in Watson–Crick conformation also with the protein bound [21,22]. Among the imino tautomers of the G:XC pairs, only the lower oxidation forms, IHC and IMC, experience a lowering of the relative free energies of the partially open/partially flipped state. That is, the conformation that likely plays a crucial role in the recognition of mispairs becomes more favourable upon complexation only for the *non*-target bases of TDG.

There are no direct interactions, such as hydrogen bonds, between the TDG protein and the G:XC pair or, for that matter, to other DNA residues, that could explain the observation of a stabilised partially open/partially flipped state in the imino tautomers. The only direct hydrogen bond between the TDG protein and the XC base is to the O15 (or O16) atom of the oxidised methyl group. But first, the probability for this hydrogen bond is very low and second, it is observed for both tautomeric forms of carboxyl-cytosine and IHC. Un-oxidised imino methyl-cytosine, IMC, whose more open conformation is most stabilised by complexation to TDG, lacks the oxygen atom in question and cannot form such an interaction.

However, only the IMC and the IHC systems populate conformations in which the imino proton is oriented in such a way that likely favourable contacts of Lys201 with the N4 atom of the XC base are possible. In the IFC and ICC systems, the higher oxidation level and hence the higher negative charge favours an orientation of the N4 atom towards the oxygen atom(s) of the formyl and carboxyl group, respectively. One can thus argue that closer interaction with Lys201 requires either a more open state, so as to allow the protein residue to come closer to the N4 atom, or a sufficiently polarised oxygen atom as in a formyl or carboxyl group. With two such oxygen atoms and a full negative charge, carboxyl-cytosine does not even need a more open conformation for stronger interactions, explaining the smallest stabilisation effect of a more open state upon complexation of ICC.

Moreover, the unoxidised methyl group is smaller than the higher oxidised forms and reduced sterical demands may be a simple explanation why IMC has the highest chance to be in a more open/more flipped state. In the case of methyl-cytosine, there are strong hydrogen bonds between the DNA residues next to the lesion and protein residues Cys233 and mainly Lys232 that are only observed for the imino tautomer, IMC. It is interesting to note that these hydrogen bonds have also been observed in earlier simulations [11,22] of DNA with G:MC and DNA with G:T complexed to TDG in both intrahelical and extrahelical conformation.

Whereas all these interactions indicate the imino tautomer IMC to be more favourable in the complex than in free DNA, the computed binding affinities point in the opposite direction. The only interaction that is diminished upon complexation is the water-mediated hydrogen bond between XC and G, while hydrogen bonds of XC (by the N4 atom) and of the complementary G17 (by its O6 atom) with water have comparable probabilities for the free and complexed IMC system. This can be interpreted as the G:IMC pair becoming even more ‘wobbly’, and occasionally too much opened for a water molecule to wedge in between. This may be the only conformational freedom gained whereas all the ‘stabilising’ contacts with the protein likely come on the expense of entropy and thus lead to an, in total, higher relative free energy.

For hydroxymethyl-cytosine, in contrast, the calculated relative binding affinities show a complexed imino tautomer to be slightly preferable over an uncomplexed IHC, in agreement with the observed stabilisation of the more open state upon complexation to TDG. For the formyl-cytosine, the imino tautomer IFC is also less favourable in the complex than in free DNA. Hydrogen bond interactions with the protein, within the DNA, and even with water are comparable for free and complexed IFC. Higher steric demands than the unmodified methyl group render the formyl group more unlikely to populate the more open wobble state. That aside, the unfavourable binding affinity to the protein for the imino tautomer IFC, may also be attributed to entropic considerations. In the complexes, the DNA fluctuates less than in the free form, and is, moreover, bent at the lesion site. Both effects are naturally due to interactions with the protein, and stabilised by for example, Arg275 wedging into the DNA groove (see Appendix A). Our earlier findings of TDG complexed to mispaired, but intrahelical, G:T, and the present study suggest partially open wobble pair conformations to be favoured by the protein and hence more stabilised. It appears that for the imino tautomers IFC and IMC this stabilisation cannot outperform the loss in entropy in these tighter complexes. For the imino tautomer ICC both effects may just balance, resulting in negligible differences of binding affinities, and for the IHC forms, the (tighter) complexation is slightly favourable.

Even the unfavourable relative binding affinities of ∼2 kcal/mol render the imino tautomers to be not much less likely in the complexes than in the free DNA. Given, however, that imino tautomers are hardly observed in free DNA in water [27,28], there is little chance for DNA with G:XC lesions in imino form to bind to the TDG protein. A transient transition from amino to imino tautomer in the complexed DNA cannot be ruled out completely, though. The subsequent base extrusion is likely considerably faster for imino than for amino tautomers, such that even the small amounts of TDG-DNA complexes with imino tautomers can significantly contribute to the formation of the extrahelical state. A step-wise binding and recognition process in which imino forms play a role could look like this: first, DNA with G:XC as amino tautomer is bound, then proton transfer in the complex generates the imino form, which is subsequently extruded. Finally, the base is, as imino or amino tautomer, expelled. Our findings of a relatively more stabilised partially open, and hence closer to extrusion, conformation for the imino tautomers of the non-cognate G:XC systems, hydroxymethyl- and methyl-cytosine, renders such a recognition mechanism favouring the wrong, that is, the non-target bases. If, however, the proton transfer step has a much lower barrier in the cognate, formyl- and carboxyl-cytosine systems than in the non-cognate systems, their imino forms, ICC and IFC, have a higher probability to be formed (in the complex) and these bases would be flipped and then excised more easily than the non-target bases. Taken together, the imino forms of the oxidised methyl-cytosines are likely not decisive for recognition by TDG upon binding to the DNA.

## 5. Conclusions

Our molecular dynamics simulations of DNA carrying different forms of oxidised methyl-cytosine in free and complexed form clearly show that the wobble conformation of the imino tautomers is maintained upon complexation to the TDG protein. As opposed to the amino tautomers, the imino tautomers exhibit two conformational states with respect to opening and base flip (extrusion). The second, less favourable, more open state of the imino tautomers is more stabilised by the TDG protein in the non-cognate G:XC systems—IHC and IMC—than in the cognate ones—IFC and ICC. This is in contrast to the relative binding affinities calculated for the amino and imino tautomers, which suggest that the IMC tautomer is less favoured in the complex than in free DNA.

According to the unfavourable binding affinities and the lack of an obvious stabilisation of the complexes of the imino tautomers IFC and ICC over IMC and IHC, the imino forms are unlikely to play an important role in substrate recognition by TDG. The amino tautomers, however, do not exhibit differences in their binding mode to the TDG protein that would favour one oxidised methyl-cytosine over the other. It is therefore conceivable that discrimination between the different oxidised forms of methyl-cytosine takes place after complex formation, at the stage of base extrusion (base flip) or latest at the chemical step.

## Figures and Tables

**Figure 1 molecules-26-05728-f001:**
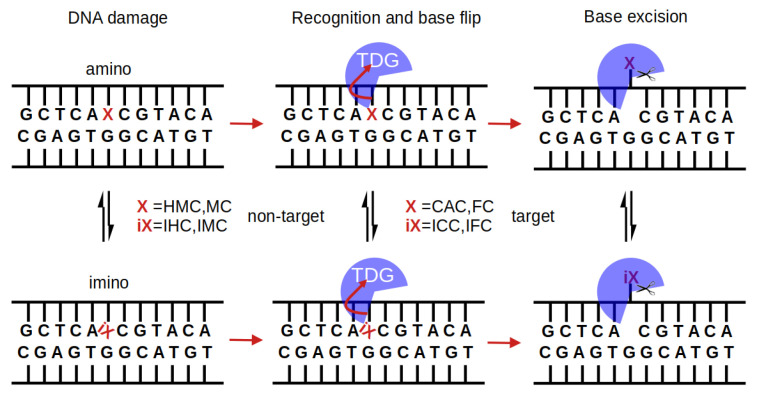
Multi-step interrogation pathway of TDG for recognition of target bases X. In their imino form (iX), the bases form a wobble pair with the G on the complementary strand (indicated by a tilted iX), which is envisaged to facilitate extrusion of the target base and flipping it into TDG’s active site [11,21,22]. This base flip site is a pre-requisite for the chemical step of base excision by glycosidic bond cleavage and crystal structures clearly show TDG-DNA complexes with the target base in an extrahelical conformation [6,7,8]. FC and CAC and their respective imino forms, IFC and ICC, are targets of TDG whereas MC and HMC (or their imino forms, IMC and IHC, respectively) are not. Discrimination between target and non-target base can in principle occur at any of the steps (protein binding, base flip, bond cleavage) and with the base in amino or imino form, though the amino form dominates in unbound DNA [6,27,28].

**Figure 2 molecules-26-05728-f002:**
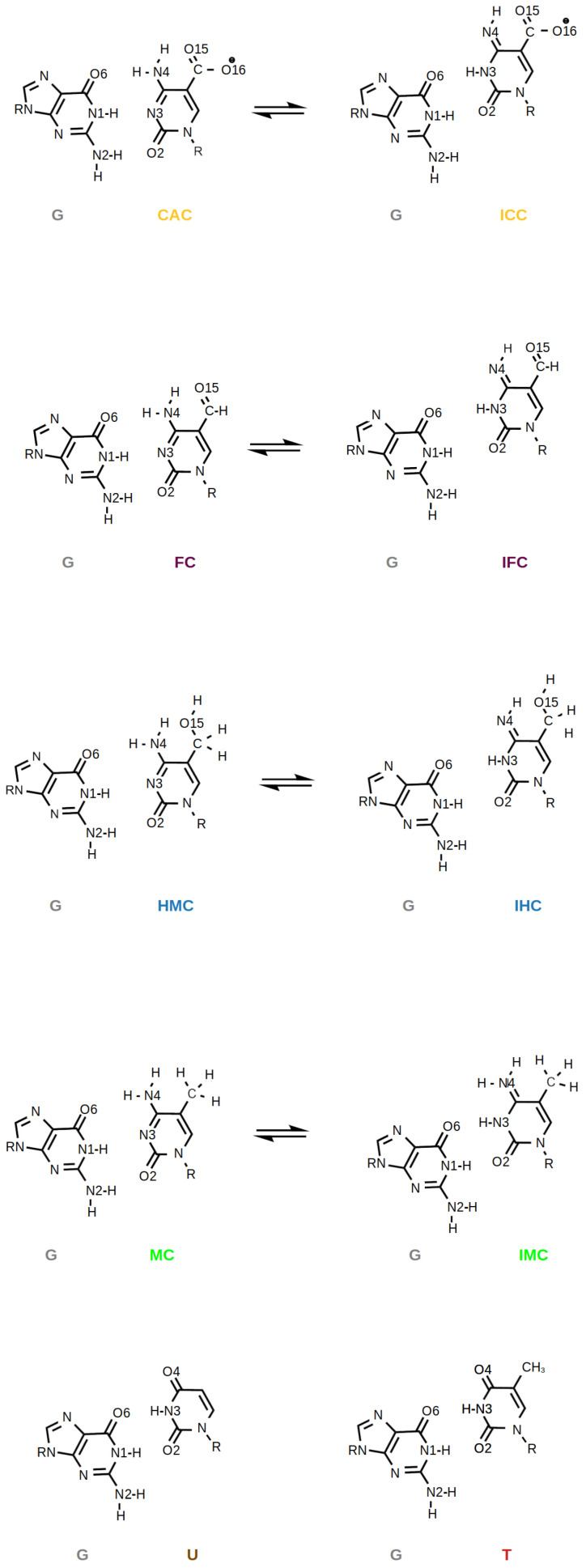
Amino and imino forms of oxidised methyl-cytosines CAC, FC, HMC, and MC paired with guanine, G, and, for comparison, the G:U and G:T mispairs.

**Figure 3 molecules-26-05728-f003:**
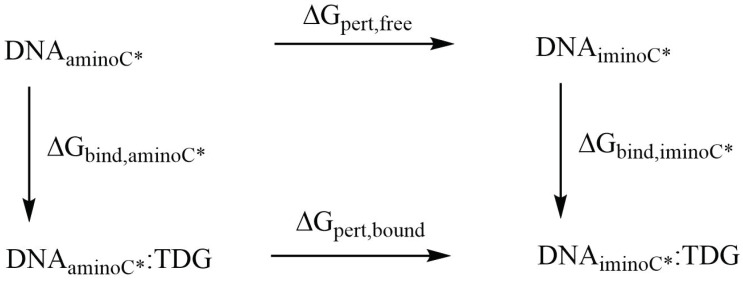
Thermodynamic cycle used for calculation of relative binding affinities.

**Figure 4 molecules-26-05728-f004:**
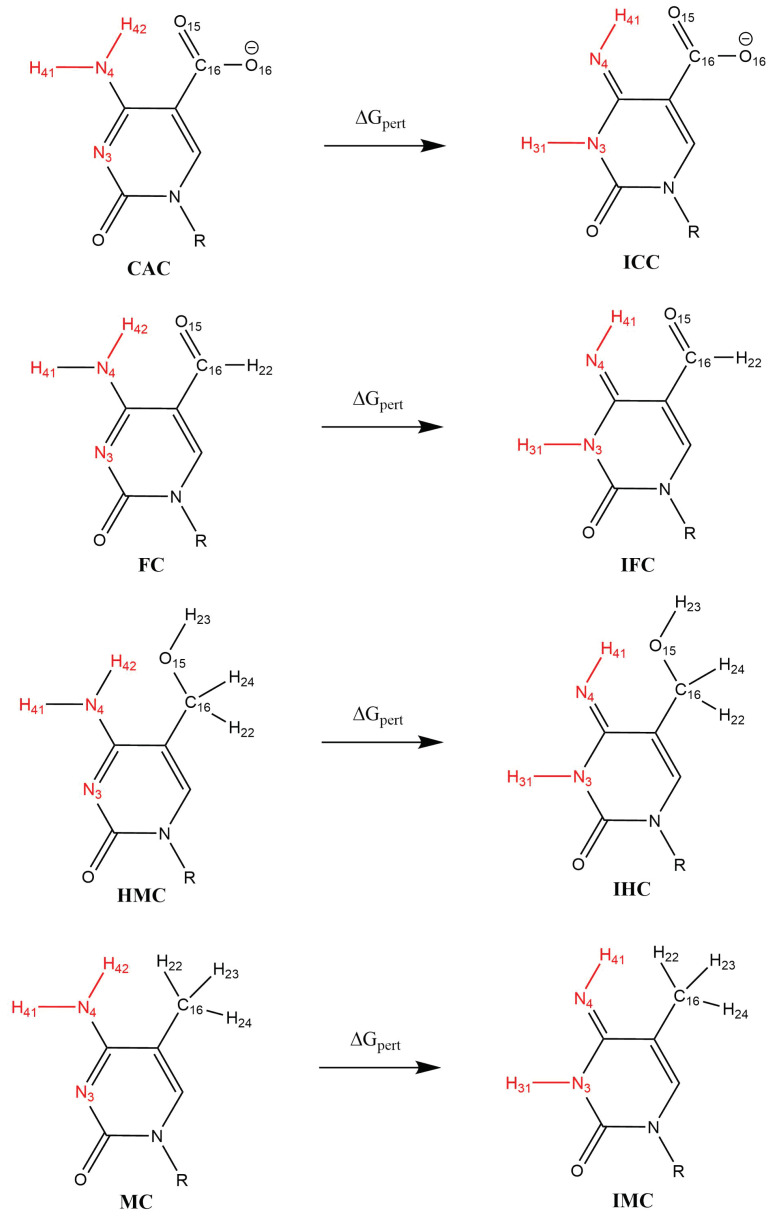
Scheme of alchemical perturbations from amino to imino forms of the oxidised methyl-cytosines. Atoms in the soft core region are shown in red.

**Figure 5 molecules-26-05728-f005:**
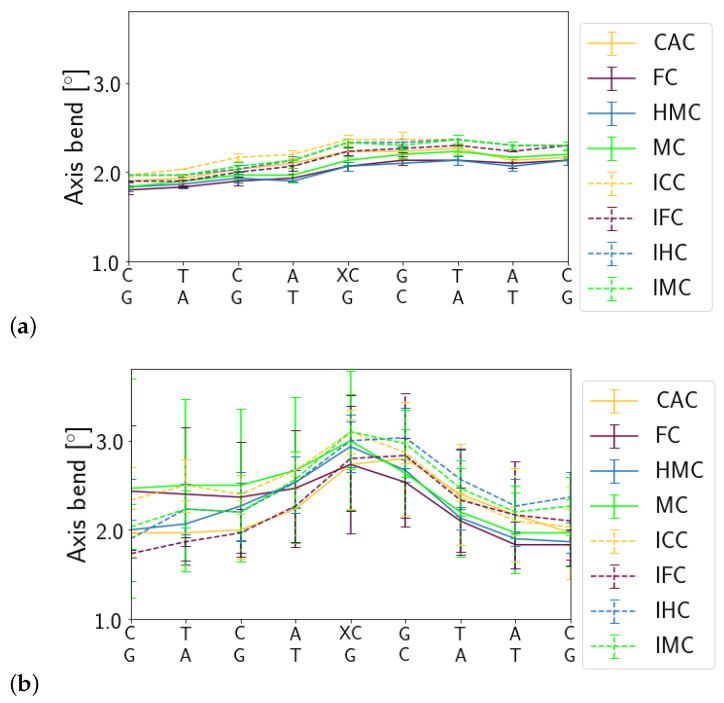
Helical axis bend of (**a**) free DNA and (**b**) complexed DNA.

**Figure 6 molecules-26-05728-f006:**
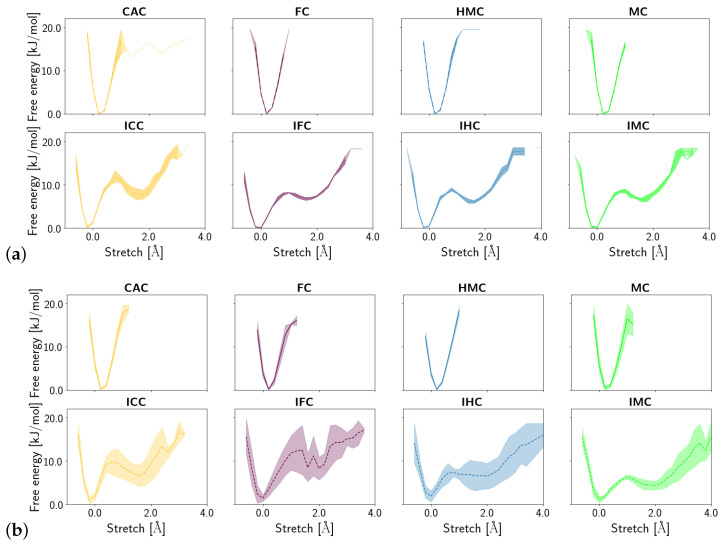
Free energy profiles of the stretch displacement of the oxidised methyl-cytosine:guanine (G:XC) pair (**a**) in the free DNA in amino form (**top**) and imino form (**bottom**) and (**b**) in the DNA complexed to TDG in amino (**top**) and imino (**bottom**) form.

**Figure 7 molecules-26-05728-f007:**
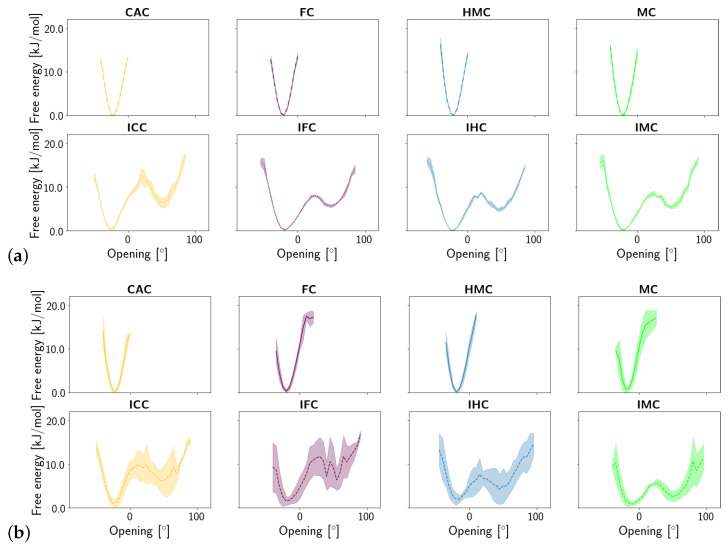
Free energy of the opening angle of the oxidised methyl-cytosine:guanine (G:XC) pair (**a**) in the free DNA in amino form (**top**) and imino form (**bottom**) and (**b**) in the DNA complexed to TDG in amino (**top**) and imino (**bottom**) form.

**Figure 8 molecules-26-05728-f008:**
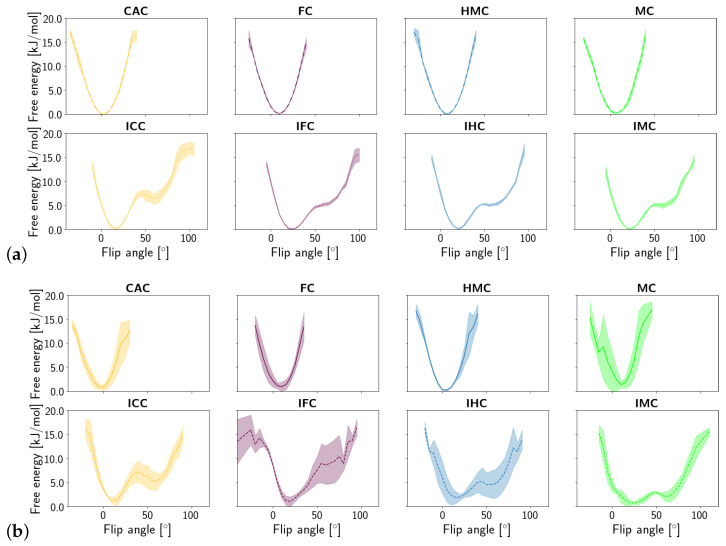
Free energy of the flip angle of the oxidised methyl-cytosine:guanine (G:XC) pair (**a**) in the free DNA in amino form (**top**) and imino form (**bottom**) and **(b)** in the DNA complexed to TDG in amino (**top**) and imino (**bottom**) form.

**Figure 9 molecules-26-05728-f009:**
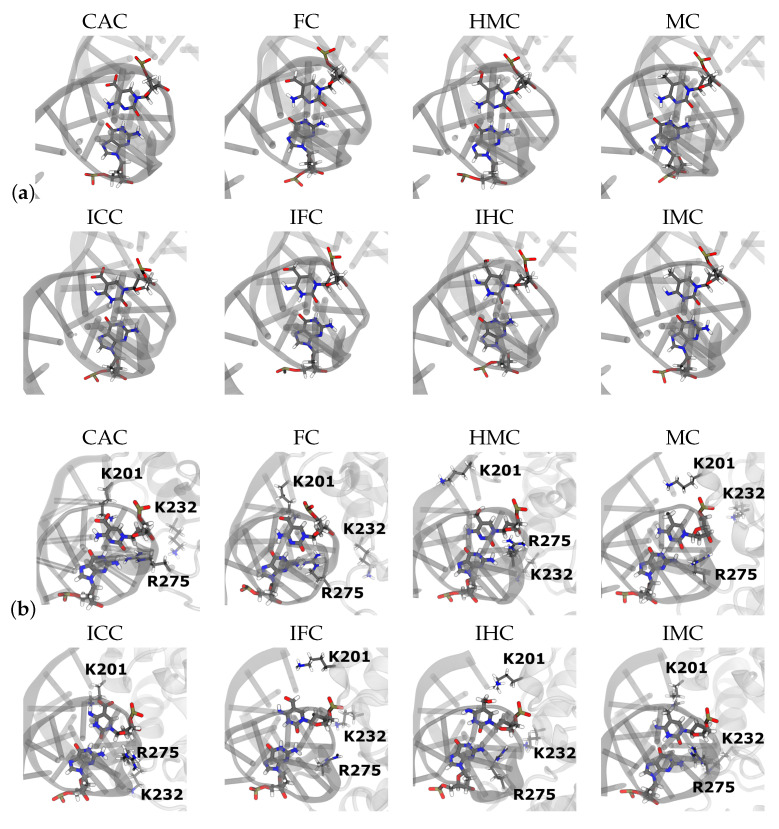
Snapshots of the G:XC pair (**a**) in free DNA and (**b**) in DNA complexed to TDG.

**Figure 10 molecules-26-05728-f010:**
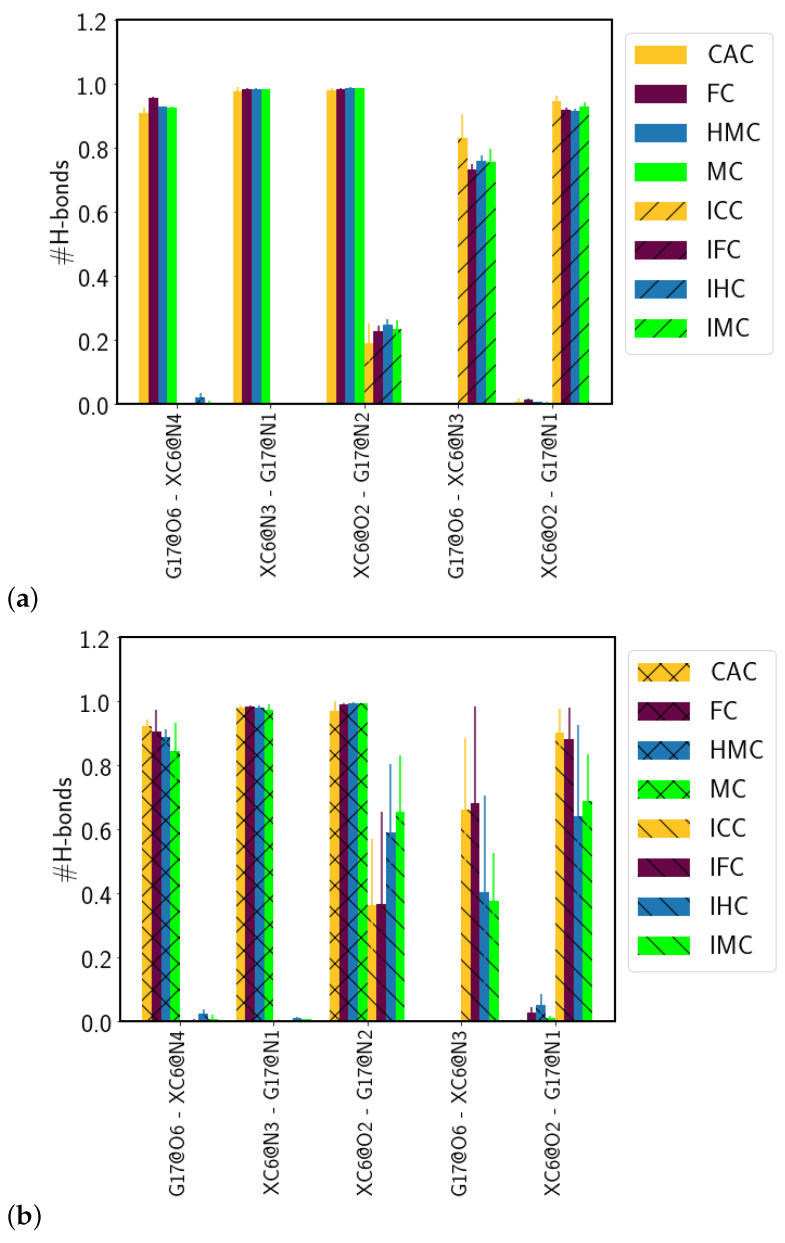
Probabilities of hydrogen bonds between the oxidised methyl-cytosine (XC) and the complementary guanine (G17) in (**a**) free DNA and (**b**) DNA complexed to TDG. Only hydrogen bonds that have a probability of at least 0.5 in at least one of the models are shown.

**Figure 11 molecules-26-05728-f011:**
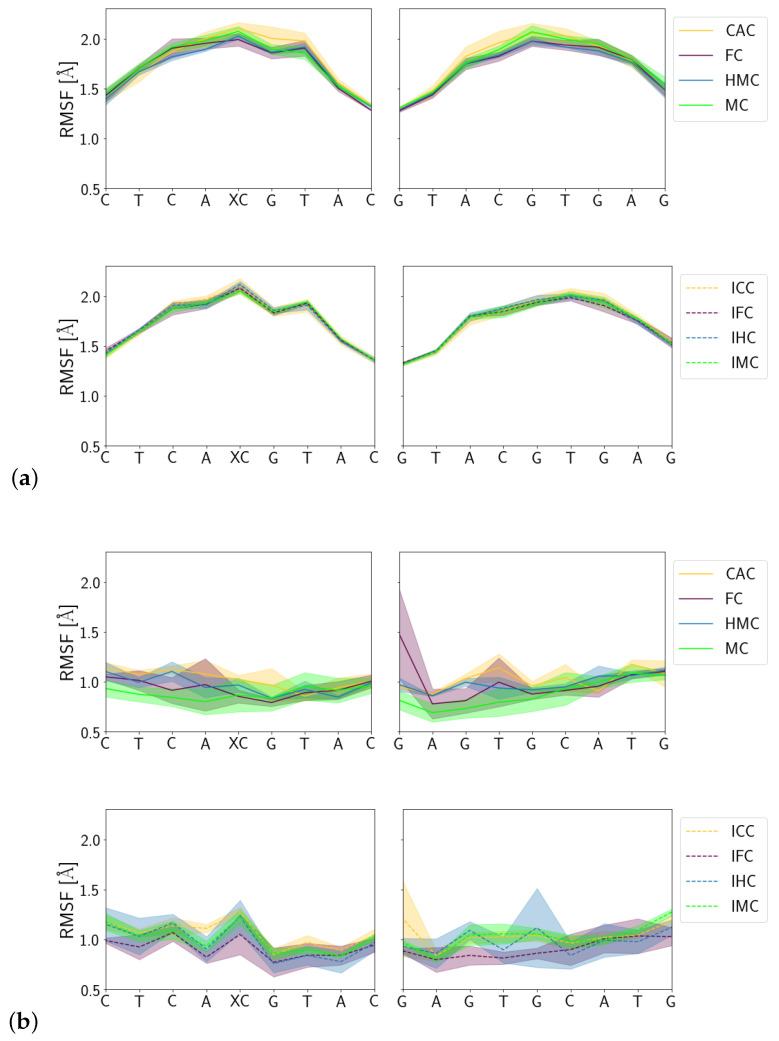
Root mean square fluctuations (RMSF) of (**a**) the free DNA and (**b**) DNA in complex with the TDG protein.

**Figure 12 molecules-26-05728-f012:**
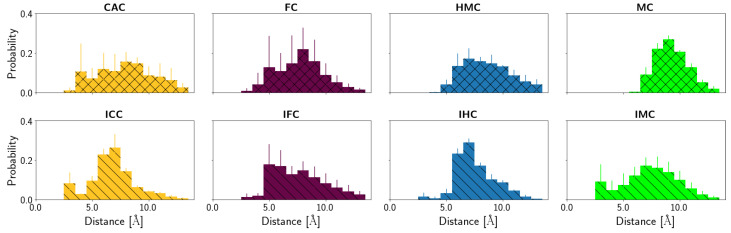
Probability distribution of the distances between Lys201 (NZ atom) and the N4 atom of the XC base.

**Figure 13 molecules-26-05728-f013:**
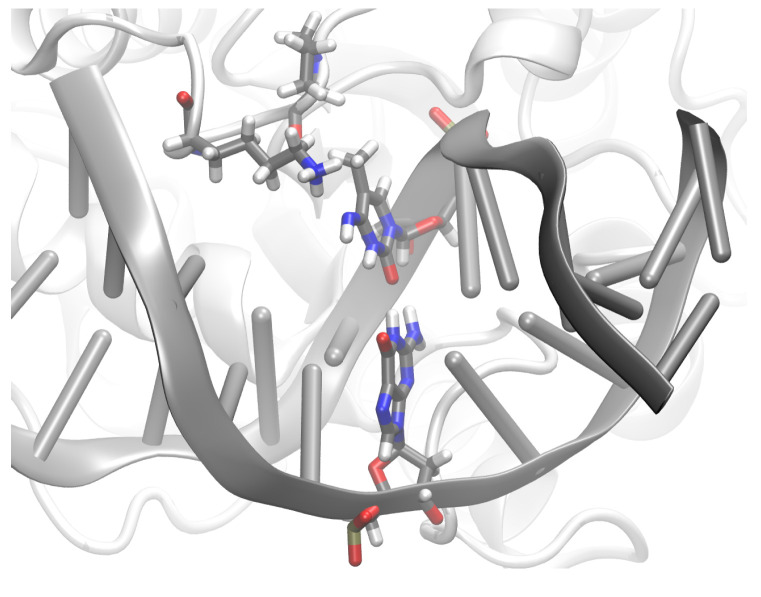
Snapshot of IMC imino methyl-cytosine in complex with TDG. Protein and DNA are represented as cartoon, the G:XC pair is shown in licorice and protein residues K201 and P202 are also shown in licorice.

**Table 1 molecules-26-05728-t001:** Probabilities of hydrogen bonds between the oxidised methyl-cytosine (XC) and the complementary guanine (G) in (a) free DNA and (b) DNA complexed to TDG. For atom labels see Figure 2. Only hydrogen bonds that have a probability of at least 0.5 in at least one of the models are listed. For hydrogen bond probabilities between all other base pairs see Appendix A.

Acc-Don	CAC	FC	HMC	MC	ICC	IFC	IHC	IMC
G17:O6-XC6:N4	0.91 ± 0.02	0.96 ± 0.00	0.93 ± 0.00	0.93 ± 0.00	–	–	–	–
XC6:N3-G17:N1	0.98 ± 0.01	0.99 ± 0.00	0.98 ± 0.00	0.98 ± 0.00	–	–	–	–
XC6:O2-G17:N2	0.98 ± 0.01	0.99 ± 0.00	0.99 ± 0.00	0.99 ± 0.00	–	–	–	–
G17:O6-XC6:N3	–	–	–	–	0.83 ± 0.08	0.73 ± 0.02	0.76 ± 0.02	0.76 ± 0.04
XC6:O2-G17:N1	–	–	–	–	0.95 ± 0.02	0.92 ± 0.01	0.92 ± 0.01	0.93 ± 0.01
**Acc-Don**	**CAC**	**FC**	**HMC**	**MC**	**ICC**	**IFC**	**IHC**	**IMC**
G17:O6-XC6:N4	0.92 ± 0.02	0.91 ± 0.07	0.89 ± 0.02	0.85 ± 0.09	–	–	–	–
XC6:N3-G17:N1	0.98 ± 0.01	0.98 ± 0.01	0.98 ± 0.01	0.97 ± 0.02	–	–	–	–
XC6:O2-G17:N2	0.97 ± 0.03	0.99 ± 0.00	1.00 ± 0.00	0.99 ± 0.00	–	–	0.59 ± 0.22	0.66 ± 0.18
G17:O6-XC6:N3	–	–	–	–	0.66 ± 0.22	0.68 ± 0.30	–	–
XC6:O2-G17:N1	–	–	–	–	0.90 ± 0.07	0.88 ± 0.10	0.64 ± 0.28	0.69 ± 0.15

**Table 2 molecules-26-05728-t002:** Probability of water-mediated hydrogen bonds between the guanine (G17) and the oxidised methyl-cytosine (XC).

	Free DNA	Complex
CAC	0.00 ± 0.00	0.01 ± 0.01
FC	0.00 ± 0.00	0.00 ± 0.00
HMC	0.00 ± 0.00	0.00 ± 0.00
MC	0.00 ± 0.00	0.00 ± 0.00
ICC	0.61 ± 0.02	0.35 ± 0.20
IFC	0.64 ± 0.00	0.50 ± 0.19
IHC	0.62 ± 0.01	0.48 ± 0.09
IMC	0.64 ± 0.01	0.52 ± 0.12

**Table 3 molecules-26-05728-t003:** Hydrogen bonds of the bases in the G:XC pair with water top: in free DNA and bottom: in DNA complexed to TDG. For atom labels see Figure 2. Note that probabilities larger than one correspond to more than one hydrogen bond formed simultaneously.

Acc-Don	CAC	FC	HMC	MC	ICC	IFC	IHC	IMC
G17:N3-W	0.65 ± 0.01	0.62 ± 0.01	0.62 ± 0.01	0.61 ± 0.02	0.66 ± 0.01	0.61 ± 0.01	0.63 ± 0.01	0.62 ± 0.01
G17:N7-W	0.81 ± 0.00	0.81 ± 0.01	0.81 ± 0.01	0.82 ± 0.00	0.85 ± 0.00	0.81 ± 0.00	0.76 ± 0.02	0.80 ± 0.01
G17:O6-W	0.97 ± 0.02	0.88 ± 0.01	0.94 ± 0.02	0.95 ± 0.01	–	0.73 ± 0.03	0.80 ± 0.09	0.68 ± 0.03
XC6:O15-W	1.79 ± 0.05	0.95 ± 0.01	0.76 ± 0.01	–	1.89 ± 0.04	0.95 ± 0.01	0.68 ± 0.01	–
XC6:O16-W	1.83 ± 0.06	–	–	–	2.09 ± 0.04	–	–	–
XC6:O2-W	0.86 ± 0.02	0.79 ± 0.01	0.86 ± 0.00	0.88 ± 0.01	0.72 ± 0.07	–	–	–
XC6:N4-W	–	–	–	–	–	0.89 ± 0.01	1.06 ± 0.15	0.94 ± 0.07
W-XC6:O15	–	–	0.84 ± 0.01	–	–	–	0.81 ± 0.00	–
**Acc-Don**	**CAC**	**FC**	**HMC**	**MC**	**ICC**	**IFC**	**IHC**	**IMC**
G17:N3-W	0.63 ± 0.11	0.52 ± 0.30	0.68 ± 0.13	0.53 ± 0.20	–	0.52 ± 0.19	–	0.55 ± 0.32
G17:N7-W	0.77 ± 0.02	0.80 ± 0.01	0.82 ± 0.04	0.86 ± 0.11	0.81 ± 0.01	0.79 ± 0.03	0.82 ± 0.05	0.80 ± 0.02
G17:O6-W	0.91 ± 0.06	0.89 ± 0.04	0.92 ± 0.05	1.02 ± 0.16	–	0.74 ± 0.23	1.06 ± 0.34	1.18 ± 0.37
XC6:O15-W	1.68 ± 0.28	0.94 ± 0.10	0.59 ± 0.10	–	1.67 ± 0.18	0.75 ± 0.18	–	–
XC6:O16-W	1.57 ± 0.20	–	–	–	1.79 ± 0.09	–	–	–
XC6:O2-W	0.68 ± 0.27	–	–	–	–	–	–	–
XC6:N4-W	–	–	–	–	–	0.87 ± 0.13	1.14 ± 0.16	1.08 ± 0.14
W-XC6:O15	–	–	0.86 ± 0.03	–	–	–	0.84 ± 0.04	–

**Table 4 molecules-26-05728-t004:** Hydrogen bond probabilities between the TDG protein and the oxidised methyl-cytosine XC. Only hydrogen bonds with a probability of at least 0.2 in at least one of the systems are shown. The hydrogen bond with the oxygen atom of the methyl group is highlighted in bold.

Acc-Don	CAC	FC	HMC	MC	ICC	IFC	IHC	IMC
**XC6:O15-LYS201:NZ**	0.24 ± 0.27	–	–	–	0.14 ± 0.17	–	0.17 ± 0.14	–
XC6:O3’-ARG275:NH	0.21 ± 0.19	–	–	–	–	–	–	–
XC6:OP-SER200:N	–	0.47 ± 0.43	0.27 ± 0.45	0.82 ± 0.06	–	0.28 ± 0.49	0.54 ± 0.47	0.32 ± 0.37
XC6:OP-SER200:OG	–	0.32 ± 0.50	0.19 ± 0.32	0.87 ± 0.11	–	0.17 ± 0.30	0.38 ± 0.39	0.22 ± 0.33
XC6:OP-LYS201:N	–	0.24 ± 0.42	–	0.40 ± 0.30	–	–	0.34 ± 0.30	–
XC6:OP-LEU143:N	–	–	0.43 ± 0.37	–	–	0.53 ± 0.46	0.12 ± 0.22	0.29 ± 0.38
XC6:OP-MET144:N	–	–	0.25 ± 0.40	–	–	–	–	–
XC6:OP-GLY199:N	–	–	0.22 ± 0.38	–	–	–	0.13 ± 0.22	–

**Table 5 molecules-26-05728-t005:** Probabilities of hydrogen bonds between the protein and the DNA but not the G:XC pair. Only hydrogen bonds that have a probability of at least 0.5 in at least one of the models are listed.

Acc-Don	CAC	FC	HMC	MC	ICC	IFC	IHC	IMC
G7:OP-SER273:OG	–	–	0.60 ± 0.24	–	–	–	–	–
T8:OP-LYS232:N	–	–	0.54 ± 0.46	–	0.91 ± 0.06	–	0.79 ± 0.15	0.87 ± 0.04
T8:OP-CYS233:N	–	–	0.62 ± 0.54	–	0.88 ± 0.07	–	0.61 ± 0.53	0.92 ± 0.01
T8:OP-SER271:N	–	–	0.56 ± 0.35	–	–	–	0.73 ± 0.37	–
A9:OP-LYS232:NZ	–	–	0.85 ± 0.01	–	0.59 ± 0.23	0.57 ± 0.36	0.86 ± 0.03	0.75 ± 0.14
T18:O2-ARG275:NH	0.60 ± 0.71	1.13 ± 0.75	1.29 ± 0.58	0.96 ± 0.89	–	1.05 ± 0.74	1.07 ± 0.58	0.58 ± 0.39
G19:N3-ARG275:NH	–	–	–	–	–	0.70 ± 0.35	–	–

**Table 6 molecules-26-05728-t006:** Relative binding affinities [kcal/mol] of DNA with oxidised methyl-cytosine in amino and imino form.

XC	$\Delta \Delta G=\Delta G_{\mathrm{bound}}^{\mathrm{pert}}-\Delta G_{\mathrm{free}}^{\mathrm{pert}}$
CAC→ICC	0.17 ± 1.10
FC→IFC	2.26 ± 0.23
HMC→IHC	−0.65 ± 0.43
MC→IMC	2.37 ± 0.26

## Data Availability

Data is available on request.

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
