# Peer review of "Interaction of Thymine DNA Glycosylase with Oxidised 5-Methyl-cytosines in Their Amino- and Imino-Forms"

_molecules, 2021, doi:10.3390/molecules26195728_

Round 1
Reviewer 1 Report
In this paper the interaction of thymine DNA glycosylase with oxidized 5-methyl-cytosine in their amino- and imino-forms, the former in a Watson-Crick conformation and the latter in a wobble one, are studied by means of Molecular Dynamics Simulations. It is shown that mispaired uracil and thymine present a wobble conformation. The comparison between free DNA and complexed DNA allows to obtain several interesting results. So, the paper is suitable for publication in the special issue of Molecules, DNA Damage and Repair.
Nevertheless, any effort to improve the presentation of the paper would be welcome. In particular:
- In the introduction, perhaps, a cartoon presenting the multi-step mechanism of repair process, would help to understand the goal of the paper.
- Many figures are so small that it is difficult to read the labels.
- In figure 1, the drawings are presented in inverse order to the one mentioned in the figure caption.
- To recognize the hydrogen bonds in tables 1 and 3, atoms numeration in figure 1 is required.
- In page 5, figure 12 is mentioned before figure 2. So, figure 12 must be figure 2 and the numeration of figures 2-11 has to be changed.
Author Response
Please attachment
Reviewer 2 Report
The manuscript titled "Interaction of Thymine DNA Glycosylase with Oxidised 5-Methyl-Cytosines in their Amino- and Imino-Forms" describes a molecular dynamics study of interaction between TDG and DNA bases.
The work new been conducted according with the method, well described.
It is not clear what is the DNA template used for this? Where is the species, these was a comparison?
The images are not clear, and some docking simulations could be useful to undestand the correct sense of the interactions.
Nevertheless, a battere explanation of the rationale must be inserted.
Round 2
Reviewer 2 Report
the manuscript has been revised according with my and other reviewers suggestions. It is now more clear, well constructed and informative for molecular biochemists but also physiologists and medicinal chemists. It is now ready for publication in Molecules.